# Novel 7-Chloro-4-aminoquinoline-benzimidazole Hybrids as Inhibitors of Cancer Cells Growth: Synthesis, Antiproliferative Activity, in Silico ADME Predictions, and Docking

**DOI:** 10.3390/molecules28020540

**Published:** 2023-01-05

**Authors:** Luka Krstulović, Marijana Leventić, Vesna Rastija, Kristina Starčević, Maja Jirouš, Ivana Janić, Maja Karnaš, Kornelija Lasić, Miroslav Bajić, Ljubica Glavaš-Obrovac

**Affiliations:** 1Department of Chemistry and Biochemistry, Faculty of Veterinary Medicine, University of Zagreb, HR-10000 Zagreb, Croatia; 2Department of Medicinal Chemistry, Biochemistry and Clinical Chemistry, Faculty of Medicine Osijek, Josip Juraj Strossmayer University of Osijek, HR-31000 Osijek, Croatia; 3Department of Agroecology and Environmental Protection, Faculty of Agrobiotechnical Sciences Osijek, Josip Juraj Strossmayer University of Osijek, HR-31000 Osijek, Croatia; 4R&D, Pliva Croatia Ltd., TEVA Group Member, HR-10000 Zagreb, Croatia

**Keywords:** aminoquinoline-benzimidazole hybrids, anticancer, in silico ADME, docking, cell cycle perturbation, mitochondrial membrane potential, apoptosis

## Abstract

In this study, new 7-chloro-4-aminoquinoline-benzimidazole compounds were synthesized and characterized by NMR, MS, and elemental analysis. These novel hybrids differ in the type of linker and in the substituent on the benzimidazole moiety. Their antiproliferative activities were evaluated on one non-tumor (MDCK1) and seven selected tumor (CaCo-2, MCF-7, CCRF-CEM, Hut78, THP-1, and Raji) cell lines by MTT test and flow cytometry analysis. The compounds with different types of linkers and an unsubstituted benzimidazole ring, **5d**, **8d**, and **12d,** showed strong cytotoxic activity (the GI_50_ ranged from 0.4 to 8 µM) and effectively suppressed the cell cycle progression in the leukemia and lymphoma cells. After 24 h of treatment, compounds **5d** and **12d** induced the disruption of the mitochondrial membrane potential as well as apoptosis in HuT78 cells. The drug-like properties and bioavailability of the compounds were calculated using the Swiss ADME web tool, and a molecular docking study was performed on tyrosine-protein kinase c-Src (PDB: 3G6H). Compound **12d** showed good solubility and permeability and bound to c-Src with an energy of −119.99 kcal/mol, forming hydrogen bonds with Glu310 and Asp404 in the active site and other residues with van der Waals interactions. The results suggest that compound **12d** could be a leading compound in the further design of effective antitumor drugs.

## 1. Introduction

Cancer is a leading disease worldwide, and the projections indicate that the global incidence of cancer will continue to rise [1,2]. The conventional anti-cancer agents are losing their efficiency [3], highlighting the need to introduce new anti-cancer entities [4,5,6]. One strategy to address the issues with existing chemotherapeutics is molecular hybridization: a strategy that generates more efficient therapeutics by combining two (or more) pharmacophores or two (or more) entire drugs in a single hybrid molecule that addresses one or multiple targets [7,8,9]. The presence of two or more pharmacophores in a single unit leads to a pharmacological potency greater than the sum of each individual moiety’s potency and may have the potential to overcome drug resistance [10]. This approach has prompted us to synthesize and evaluate new hybrid molecules composed of 7-chloroquinoline and benzimidazole structures.

Benzimidazoles are nitrogen-containing heterocyclic aromatic compounds that have a wide variety of chemically therapeutic functions and a significant biological activity profile [11,12]. Over the years, extensive research has shown that the benzimidazole nucleus is also an important core of various agents that are approved or under clinical evaluation as anti-cancer therapies (Figure 1). The anticancer therapeutic Bendamustine, shown in Figure 1, is a multifunctional DNA cross-linking alkylating agent composed of a mechlorethamine group, a butyric acid side chain, and a benzimidazole. It has demonstrated clinical activity against hematologic malignancies when used as a monotherapy or in combination with other chemotherapeutics. It is currently indicated for the treatment of chronic lymphocytic leukemia (CLL) and indolent B cell non-Hodgkin’s lymphoma (NHL) [13,14]. The next important antitumor drug with a benzimidazole moiety is Binimetinib (Figure 1), whose mechanism of action is related to the inhibition of mitogen-activated protein kinases (MEK-1 and 2). Binimetinib has been approved by the FDA for the treatment of melanoma with BRAF mutations [15,16] and has been under clinical investigation as part of a combined therapy for the treatment of colorectal cancers [17,18]. Furthermore, Crenolanib is a benzimidazole that contains a quinoline moiety, among others. It acts as an inhibitor of FMS-like tyrosine kinase 3 (FLT3) and has been evaluated in various clinical trials against acute myeloid leukemia [19].

On the other hand, the quinoline fragment is a part of various biologically active compounds with antiviral, antimicrobial, antiparasitic, and anti-fungal activity [20]. There have also been numerous publications describing the anti-cancer activity of quinoline-based derivatives and their different mechanisms of action [21,22]. The quinolone anti-cancer therapeutic Lenvatinib, shown in Figure 1, is a kinase inhibitor against various growth factor receptors and is approved for the treatment of thyroid and hepatocellular cancer [23,24]. Bosutinib is another quinoline-based anti-cancer agent; it is a Bcr-Abl kinase inhibitor used to treat chronic myelogenous leukemia [25]. In recent years, established antimalarial 7-chloroquinolines such as chloroquine and hydroxychloroquine have been investigated in clinical trials as promising anticancer drugs [26,27].

Considering the antiproliferative activity of the compounds with the aforementioned moieties and the continuation of our work on quinoline-benzimidazole hybrids [28,29], the aim of this study is to identify the physicochemical properties and structural attributes for the anticancer activity, as well as the interactions of 7-chloroquinoline-benzimidazole hybrids on the c-SRC by molecular docking, based on the binding mode of the DSA inhibitor. In addition, in silico physicochemical and pharmacokinetic/ADMET (absorption, distribution, metabolism, excretion, and toxicity) studies were undertaken in order to theoretically predict their behavior as drug candidates.

## 2. Results and Discussion

### 2.1. Chemistry

#### Design and Synthesis of Novel Benzimidazole Derivatives

The synthesis of novel benzimidazole derivatives of 7-chloro-4-aminoquinoline **5a–e**, **8a–e**, and **12a–d** was carried out as shown in Figure 1 and Figure 2. With the aim of exploring the influence of the linkers between two pharmacophores, as well as the effect of the substituent at the benzimidazole ring, benzimidazoles with three different amidine substituents or chlorine or without a substituent at the C-5 position and 7-chloroquinoline were joined through three different linkers: 3-phenyl, 4-phenyl piperazine, and ethyl benzamidyl.

Alcohol **2** and aldehyde **3** were prepared using the methods reported by Medlen et al. [30]. 7-chloroquinoline and 3-aminobenzyl alcohol were refluxed in ethanol to yield (3-(7-chloroquinolin-4-ylamino) phenyl)methanol **2**. The latter compound was transferred to 3-(7-chloroquinolin-4-ylamino)benzaldehyde **3** by stirring in DMSO with trimethylamine and sulfur trioxide complex in pyridine at room temperature. Quinoline benzimidazoles **5a–e** were prepared by employing a condensation reaction between aldehyde **3** and benzene-1,2-diamines **4a–e** in the presence of sodium metabisulfite Na_2_S_2_O_5_ in DMSO at 165 °C [31]. The synthesis of the precursors **4a–c** was reported earlier [32,33]. All the analytical data of the earlier synthesized compounds correspond with the information previously given and will not be shown in this paper. The synthesis of 7-chloro-4-(piperazin-1-yl)quinoline **6** was also reported earlier [34]. The latter compound, via the nucleophilic substitution of fluorine in 4-fluorobenzaldehyde, resulted in 4-(4-(7-chloroquinolin-4-yl)piperazin-1-yl)benzaldehyde **7**. Quinoline benzimidazoles **8a–e** were synthesized by the condensation of aldehyde **7** and diamine precursors **4a–e** with Na_2_S_2_O_5_.

Figure 2 illustrates the synthetic route for the compounds with the ethyl benzamide linker **12a–d**. Precursors with a carboxylic moiety **10a–d** were prepared by the aforementioned condensation with Na_2_S_2_O_5_ from 4-formylbenzoic acid (**9**) and diamines **4a–d**. The previously reported [35] *N*-(aminomethyl)-7-chloroquinolin-4-amine **11** was linked with the precursors **10a–d** in the presence of trimethylamine, 1-hydroxybenzotriazole (HOBt), and *N*-(3-Dimethylaminopropyl)-*N*′-ethylcarbodiimide hydrochloride (EDCxHCl) in dichloromethane, a method previously reported by Pešić et al. [36]. We were not able to synthesize a chlorine derivative from this series. Although we prepared a chlorine precursor with a carboxylic moiety, the latter compound in the reaction with amine **11** did not yield the desired product. The NMR spectra of the material isolated from this reaction showed signals which correspond to both the starting amine **11** and the starting chloro benzimidazole benzoic acid derivative. In the NMR spectra, we did not find signals which would clearly indicate that the reaction actually took place. We tried using column cromatography to seperate the reaction mixture components, but we did not isolate any component which did not correspond to the starting compounds. Our attempts to modify the reaction conditions, which were employed for the synthesis of the corresponding compounds **12a–d**, were also unsuccessful. The only conclusion we draw is that under these conditions the chlorine group, on the benzimidazole moiety, had a deactivating effect.

In our previous efforts to synthesize new benzimidazoles by the condensation of benzene-1,2-diamines and aryl aldehydes, we readily employed a method with sodium bisulfite NaHSO_3_^,^ [37,38]. Utilizing Na_2_S_2_O_5_ in the present paper was advantageous because of the shorter reaction time (15 min. vs. 6 h for NaHSO_3_). In addition, in the case of the compounds **8a–e** and **10a–d**, the reactions with Na_2_S_2_O_5_ yielded pure products with no need for further purification; this was not the case when we tried to prepare corresponding compounds with NaHSO_3_.

### 2.2. Biological Activity

#### 2.2.1. Evaluation of Antiproliferative Activity of the Novel Compounds

The newly synthesized benzimidazole derivatives of 7-chloro-4-aminoquinoline were tested for their effects on the growth of the non-tumor MDCK1 (Madine–Darby canine kidney) cell line, the leukemia and lymphoma cell lines: CCRF-CEM (human acute lymphoblastic *leukemia*), Hut78 (T-cell lymphoma), THP-1 (acute monocytic leukemia), Burkitt lymphoma (Raji), and the carcinoma cell lines: HeLa (human cervical adenocarcinoma), CaCo-2 (human colorectal adenocarcinoma), and MCF-7 (human breast adenocarcinoma). The obtained results, presented as the concentration achieving 50% of cell growth inhibition (GI_50_ value), show that the investigated compounds differently influenced tumor cell growth, depending on the cell line and on the dose applied.

As shown in Table 1, the GI_50_ values for all the investigated compounds ranged from 0.6 to more than 100 μM. The carcinoma cell lines were more resistant to the investigated compounds compared to the leukemia and lymphoma cells. The novel quinoline benzimidazoles **5a–c**, in which the pharmacophores are connected with a 3-phenyl linker and the benzimidazole ring carries one of the three different amidine substituents (Figure 1), exhibited insignificant inhibitory potential against normal and tumor cells. Compounds **5d** (containing an unsubstituted benzimidazole ring) and **5e** (containing a benzimidazole ring substituted with chlorine) showed mild to strong inhibitory potential against the normal (GI_50_ ranging from 11.4 to 20.4 μM) and the tumor cell lines (GI_50_ values ranging from 0.4 to 15.6 μM). In comparison to the MDCK1 and the carcinoma cell lines, compound **5d** exhibited significant antiproliferative activity against the lymphoma (Raji: GI_50_ = 4.3 μM; HuT78: GI_50_ = 0.4 μM) and leukemia cells (CCRF CEM: GI_50_ = 8.2 μM; THP1: GI_50_ = 0.6 μM). With respect to the inhibition potential of **5d** against the MDCK1 cells, the tumor selectivity index (SI) of the HuT78 cells is 51 and of the THP1 cells it is 34. The replacement of hydrogen with chlorine at the benzimidazole ring in **5e** caused strong antiproliferative activity on all the tested cell lines.

Furthermore, in the group of compounds containing the 4-phenyl piperazine linker (**8a–e**), a mild to strong, albeit not selective, antiproliferative activity was observed (Table 1). Compounds **8a–c**, in which the benzimidazole ring carries one of the three different amidine substituents (Figure 1), showed moderate effects on the growth of normal and tumor cells, with GI_50_ values ranging from 20.7 to more than 100 μM. Compound **8d**, which possesses an unsubstituted benzimidazole ring, significantly influenced the growth of the lymphoma and leukemia cells (CCRF CEM: GI_50_ = 5.0 μM, SI = 20; THP1: GI_50_ = 3.2 μM, SI = 31.5; Raji: GI_50_ = 3.8 μM, SI = 26.3; HuT78: GI_50_ = 8.1 μM, SI = 12.3). At the same time, **8d** had a negligible effect on the growth of normal and carcinoma cells, with GI_50_ values of more than 100 μM. The replacement of hydrogen with chlorine at the benzimidazole ring in **8e** caused a significant antiproliferative effect on the normal MDCK1 cells (GI_50_ = 2.7 μM). Compound **8e** expressed less cytotoxicity effects against the carcinoma cells (GI_50_ ranging from 84.0 to >100 μM) and modest effects on the growth of the leukemia and lymphoma cells (GI_50_ ranging from 13.4 to 41 μM) in comparison to the effects on the MDCK1 cells.

In the group of quinoline benzimidazoles, in which the pharmacophores are connected with the ethyl benzamidyl linker (Figure 2), a significant distinction in the antiproliferative effects between the normal and the most tumorous cell lines for compounds **12a–c** was not observed (Table 1). An exception was the HuT78 cell line, which was significantly sensitive to the effects of the compounds **12a–d** (**12a**: GI_50_ = 4.1 μM, SI = 24.4; **12b**: GI_50_ = 5.6 μM, SI = 17.9; **12c**: GI_50_ = 4.4 μM, SI = 22.7; **12d**: GI_50_ = 3.5 μM, SI = 28.5).

The antiproliferative affinities of compounds **5d** (against the THP1 and HuT 78 cells), **8d** (against the CCRF-CEM, THP1, Raji and HuT78 cells), and **12d** (against the THP1 and HuT78 cells) were significantly higher (*p* < 0.001, two-way ANOVA with Bonferroni post-test) compared to the healthy cell line, indicating therapeutic potential.

An insight into the effect of the different substituents at the benzimidazole moiety and the roles of the varied linkers is shown in Figure 2. Non-amidine compounds **5d** and **5e** with a 3-phenyl linker demonstrated the highest antiproliferative activity against the tumor cell lines, but they were not selective. The introduction of non-amidine substituents into compounds **8d** (-H), **8e** (-Cl), and **12d** (-H) resulted in a higher activity against most of the leukemia and lymphoma cell lines. Compounds **5d**, **8d**, and **12d**, which do not have a substituent at the C-5 position of the benzimidazole, demonstrated a general increase in selectivity compared to the compounds substituted with amidine.

#### 2.2.2. Cell Cycle Redistribution

The delimited cell cycle progression and the evasion of apoptosis are the common events in the development of most tumors. The cell cycle checkpoints fortify the dividing cells against DNA damage and ensure the maintenance of genomic integrity. To assess whether a cell cycle disturbance underlines the antiproliferative activity that some compounds exhibited against lymphoma and leukemia cells in micromolar and submicromolar concentrations, we tested the cell cycle distribution of the HuT-78 and THP-1 cells treated with **5d**, **8d**, and **12d**. As is shown in Figure 3, the compounds **5d** and **12d** induced a statistically significant redistribution in the cell cycles of the treated HuT78 and THP-1 cells compared to the control (untreated) cells. Compound **5d** caused a significant increase in the aggregation of cells in the subG0/G1 phase in both the treated cell lines; a significant increase in the aggregation of cells in the subG0/G1 phase (HuT78: 13 times, *p* < 0.002; THP-1: (2.9 times, *p* < 0.001); and a significant decrease in the G0/G1 phase (HuT78: 5 times, *p* < 0.00; THP-1: 4.5 times, *p* < 0.004), with a consequent increase in the G2/M phase (HuT78: 2 times, *p* < 0.003; THP-1: 62.8%, *p* < 0.034) of the cell cycle.

Furthermore, compound **12d** induced a statistically significant enrichment of the subG0/G1 fraction in the treated HuT78 cells (46 times, *p* < 0.004) and a significant decrease in the G0/G1 phase (for 63.8%, *p* < 0.004) compared to the non-treated cells. After treatment with **12d**, the THP-1 cells showed an increased percentage of cells in the subG0/G1 phase (3.4 times, *p* < 0.006), while the changes in the other phases of the cell cycle were not statistically significant. These results indicate the prevention of the entry of the treated cells into a new cell cycle and are in accordance with the results published by Zuo at al. and Zhang at al. [39,40]. The subG0/G1 arrest suggests that the initiation of cell cycle arrest may be responsible for the antiproliferative potential. The subG0/G1 peak is indicative of DNA fragmentation and suggests that the cells are dying, most likely through apoptosis. The arrest of the cell cycle in the G2/M phase in the cells treated with the various benzimidazole derivatives has also been observed in several recently published studies [41].

An increased proportion of polyploid giant cancer cells was found in the HuT78 cells treated with compounds **5d** and **12d** and in the THP-1 cells treated with compound **5d**. The increase in the number of polyploids could be explained by the cellular stress due to the activity of the applied compounds, as suggested by previous studies [42,43], or it could be the way in which the cells try to avoid the therapy [44,45].

As shown in Figure 3, no significant changes in the cell cycle were observed in the HuT78 and THP-1 cells treated with compound **8d** in comparison with the non-treated cells, indicating a different mechanism of the antiproliferative effect compared to the effects of compounds **5d** and **12d**.

#### 2.2.3. Apoptosis Induction

The induction of cell cycle arrest at a specific checkpoint and the initiation of apoptosis are mechanisms which are often used for treating cancer with cytotoxic agents [46,47,48]. To prove apoptosis as a mechanism of the treated cell death of HuT78 cells exposed to **5d** and **12d**, we used two methods for apoptosis detection: changes in the mitochondrial membrane potential (∆Ψm) and the phosphatidylserine externalization on the plasma membrane as typical features of apoptotic cell death.

Changes in the (∆Ψm) were measured using TMRE (tetramethylrhodamine, ethyl ester, and perchlorate) dye. Flow cytometry analysis showed that compound **12d** caused a statistically significant mitochondrial membrane disruption in more than 85% of the HuT78 cells. Compound **5d** also induced a change in the mitochondrial membrane potential in 23.9% of cells (Figure 4). Complex mitochondrial changes and alterations in the mitochondrial membrane potential may be an early event in the apoptotic process or, conversely, may be a consequence of the apoptotic signaling pathway [49,50]. In response to multiple distinct intracellular stress conditions, mitochondrial membranes can become permeabilized because of the pore-forming activity of the proapoptotic members of the Bcl-2 protein family. Alternatively, mitochondria can lose their structural integrity after the mitochondrial permeability transition, a phenomenon that is initiated at the mitochondrial inner membrane [50,51]. In both cases, permeabilized mitochondria allow the release of proapoptotic proteins into the cellular cytoplasm. The obtained results are consistent with the results of the previously published studies, which showed that some benzothiazole derivatives have the potential to induce the apoptosis of B and T lymphoma cells through the intrinsic pathway by the disruption of the mitochondrial membranes [52].

The results of the flow cytometry analysis of the Annexin V binding on the HuT78 cell membrane after 24 h treatment with **5d** and **12d** showed that **12d** is a potent inducer of cell death (Figure 5). After treatment with **12d**, a high percentage of cells were in the late stage of apoptosis/necrosis (83.75%), while the proportion of cells in early apoptosis was negligible (3.97%). The high proportion of cells observed in late apoptosis, 24 h after treatment, may indicate the rapid effect of **12d** on Hut78 cells, likely inducing apoptosis soon after the application.

The proapoptotic effect of compound **5d** is less dramatic compared to that of compound **12d**. After 24 h of the treatment of the HuT78 cells with this compound, about 64% of the cells were alive, 24.75% of the cells were in early apoptosis, and only 8.3% of the cells were in late apoptosis. (Figure 5).

### 2.3. Absorption, Distribution, Metabolism, Excretion (ADME), and Toxicity Properties

The pharmacokinetic behavior of a drug compound determines its fate in the organism, and it is exploited to predict the ADME properties of that drug compound [53,54]. The physicochemical profiles of potential drugs tend to take into account molecular weight, ability to form hydrogen bonds, and lipophilicity. *The octanol*-*water* partition *ratio* is a common way of expressing the lipophilicity of a compound. There are many methods for the calculation of the logarithmic value of the octanol/water partition coefficient (log*P*). The Lipinski rule uses the Moriguchi-based algorithm to calculate log*P* (MLOGP). The advantages of the Moriguchi method are the ease of programming in any language and that this method does not require a large database of parameter values [55]. The in silico ADME properties for the novel 14 compounds were investigated and are recorded in Table 2. According to the calculated ADME parameters, compound **5e**, which was found to be one of the most active compounds against cancer cell growth, could be problematic as an effective oral drug since its MLOGP is higher than 4.15 and the log S is lower than −6, which means that is highly lipophilic and poorly soluble in water. Additionally, its fraction of carbons in the sp3 hybridization (saturation) is zero since its structure is missing saturated carbon-carbon bonds [55]. Compound **5d** also exhibited a strong inhibitory potential against the growth of all the tested cancer cell lines. Although its Csp^3^ fraction is zero and the log S is <−6, other ADME properties indicate that this compound could have good oral bioavailability. Thus, it fully satisfies the Lipinski rule. Compound **12d** showed good inhibition activity against most of the cell lines. Except for the inappropriate fraction of Csp^3^, the other ADME properties suggest that molecule **12d** has good solubility and permeability and could be effective as an antitumor drug.

The pharmacokinetic behavior of a set of compounds was evaluated using the program pkCSM [56]. This program uses graph-based signatures to develop predictive models of the central ADMET properties, which are important factors to consider when developing a new drug. These properties include human intestinal absorption (HIA/% abs), volume of distribution at steady state in humans (VDss), blood–brain barrier permeability (BBB per.), CNS (central nervous system) permeability, total clearance, oral rat chronic toxicity (LOAEL), and flathead minnow toxicity. The results of this evaluation are presented in Table 3. The compounds **5a** and **8a** had the highest HIA values, at 95.43% and 95.68%, respectively. HIA is a crucial ADME property because it is one of the key steps in the transport of drugs to their targets in the body [57]. Compound **12b** has the highest VDss (0.35 log L kg^−1^). This means that a higher dose of this compound would be required to achieve a given plasma concentration. The lowest values of VDss were found for compounds **8d** and **8e** (−0.13, −0.12, respectively), which corresponds to their highest blood-brain barrier permeability (0.53, 0.49, respectively), and it corresponds to their highest CNS permeability (−0.54, −0.59, respectively). This indicated that these compounds have the possibility to easily leave the plasma and enter the extravascular compartments of the body, and after passing the blood–brain barrier, they can easily target the central nervous system [58]. Total clearance is an important pharmacokinetic parameter that describes the rate at which a drug is removed from the plasma. It is calculated by dividing the rate of drug removal from the plasma (mg/min) by the concentration of the drug in the plasma (mg/mL). In this case, the compound with the highest clearance was **5c**, at 1.01 log mL min^−1^ kg^−1^ [59]. The highest aquatic toxicity evaluated against the fish model organism flathead minnow was estimated for compound **8e** (−2.92 log mM), while the most toxic, for the rats, was shown to be compound **5b** (1.25 log mg^−1^ kgbw per day). Both of these compounds are very lipophilic and poorly soluble in water, as indicated in Table 3.

### 2.4. Molecular Docking

Tyrosine kinases are enzymes that phosphorylate tyrosine residues in specific substrates, and they are activated when a ligand binds to their extracellular domain. Inhibitors of the low molecular weight epidermal growth factor receptor (EGFR) can block extracellular signaling molecules (EGF) and prevent cell proliferation [60]. Src kinases play a crucial role in the assembly and disassembly of tight junctions in CaCo-2 and MDCK1 cell monolayers. It has been shown that the oxidative stress-induced disruption of tight junctions, which form a barrier to the diffusion of toxins, allergens, and pathogens across epithelial tissue, is mediated by the activation of c-Src [61]. The ATP binding pocket of Src is located in the cleft between the two lobes of the enzyme, the small amino-terminal (N) lobe made up of residues 267–337, and the large carboxyl-terminal (C) lobe made up of residues 341,520. The two lobes can move relative to each other, opening and closing the cleft. In the inactive conformation of the kinase, the Asp-Phe-Gly (DFG) motif is flipped by about 180° relative to the active state conformation. The flipped DFG motif moves the Asp away from the ATP binding site, leading to a “DFG”-out state. In the active conformation, the Asp side chain faces into the ATP binding pocket and coordinates with a Mg^2+^ ion, resulting in a “DFG”-in state [62,63].

As most of the compounds have shown inhibitory activity against the leukemia and lymphoma cell lines (CCRF-CEM, THP1, Raji, and HuT78), tyrosine-protein kinase c-Src (PDB: 3G6H) was chosen as a receptor for the molecular binding studies. These studies aim to predict the ability and mode of interaction of the synthesized molecules with the c-Src enzyme, and the results are compared to the docking results of the bonded ligand pyridinyl-triazine (DSA1, PDB ID: G6H) [62]. The ranking of the screened compounds based on the energies of their interactions is presented in Table 4. These molecular binding studies can provide insights into the potential of the synthesized compounds as ligands for c-Src and their potential as therapeutic agents.

The highest negative total energy (−150.08 kcal mol^−1^) overtakes compound **12c**, ahead of the original ligand, compound DSA1. Compound **12c** fits best into the active site of c-Src, as demonstrated by its significant effect on the growth of the HuT78 cell line (Table 1). However, this compound proved to be toxic to the normal, MDCK1 cells.

Compound **DSA1**, a bonded ligand (G6H), demonstrated slightly weaker binding energy (−133.74 kcal mol^−1^) than compound **12c**. These compounds have the structure scaffold of Imatinib, the effective drug for chronic myeloid leukemia (Gleevec, Novartis), which binds with a higher potency to c-Src in the DFG-flipped conformation. Compound **12b** appears in the ranking with the total energy of −122.66 kcal mol^−1^. This is consistent with the fact that compound **12b** is also the significant active compound against the leukemia and lymphoma cells: Raji and HuT78. Compound **12b** belongs to the group of compounds (**12a–d**) with an extension of the molecules’ central linker between the chloroquinoline and benzimidazole parts by the ethyl group, which contains two amino groups. This is in agreement with the previous QSAR findings on the anticancer activities of the CQArA hybrids that the 2-aminoethanol linker enhances their anticancer activity [29]. Unfortunately, compound **12b** also exhibited an inhibitory potential against the normal cell line (Table 1). One of the compounds that demonstrated the best toxicity ratio in normal and tumor cells, especially leukemia and lymphoma cells, is **12d**. This compound binds to the active site of the c-Src, releasing energy of 119.99 kcal mol^−1^ (Figure 6). The energies of the interactions between the protein residue and ligand **12d** in docked pose 1 are tabulated in Table 5. The interactions of compound **12d** with c-Src in the binding site are presented in Figure 6, which shows compound **12d** docked in the binding site of c-Src, presented as a hydrophobic surface, while Figure 7 represents the 2D interactions of the ligand **12d** with the residual amino acid of receptor c-Src. Like DS1, compound **12d** is anchored into a hydrophobic pocket composed of the side chains of Val313, Leu317, Leu322, Val377, and Tyr382 [64]. Compound **12d** forms hydrogen bonds with Glu310 (2.77 Å) and Asp404 (3.34 Å) via an amino group from the central linker, *N*-(2-aminoethyl)benzamide. The same group generates a carbon hydrogen bond with residue Thr338, which is the gatekeeper that controls the access of the inhibitor to the active site in c-Src [64]. The other key van der Waals interactions are between the quinoline ring and Val323 (3.71 Å and 4.45 Å); Met314 (4.84 and 5.02 Å); and Ala403 (4.50 Å). The phenyl ring of benzamide generates π-π interactions with Ala293 (4.43 Å) and Lys295 (5.27 Å), as well as a π-σ interaction with Val281 (5.00 Å). Significant π-π T-shaped interaction creates a benzimidazole ring with Phe405 (4.66 Å), as well as two π-π interactions with Lue273 (5.07 and 5.27 Å). For comparison, Imatinib and the inhibitor triazine derivative DSA8 form a hydrogen bond with Asp404 in the Asp-Phe-Gly (DFG) motif [62]. The results of our previous docking study of quinoline-arylamidine hybrids with c-Src revealed that the key interactions are the H-bonds between the amino groups from the 2-aminoethanol linker, Asp404 and the quinoline nitrogen atom, and Tyr382 and with Tyr340 via nitrogen from the nitrile group [29]. Apigenin glycoside isolated from *Cupressus sempervirens* creates a hydrogen bond via the hydroxyl group of a glucopyranose ring with Val402 and Val323, while the second hydroxyl group creates hydrogen bonds with Tyr382 and Asp404 [65]. A docking study of azaacridine derivatives revealed that the amido moiety also formed two hydrogen bonds with Glu310 and Asp404 and three π-π interaction bonds and one cation–π bond with Leu 273, Phe 405, Tyr 382, and Lys 295, respectively [66].

## 3. Materials and Methods

### 3.1. Chemistry

All the solvents and reagents were used without purification from commercial sources. To monitor the progress of a reaction and for comparison purposes, thin layer chromatography (TLC) was performed on pre-coated Merck silica gel 60F-254 plates using an appropriate solvent system, and the spots were detected under UV light (254 nm). For column chromatography, silica gel (Fluka, 0.063e0.2 mm) was employed; the glass column was slurry-packed under gravity. The melting points (uncorrected) were determined with the Buchi 510 melting point apparatus. The ^1^H and ^13^C NMR spectra were acquired on a Bruker 300 and 600 MHz NMR spectrometer. All the data were recorded in DMSO-d6 at 298 K. The chemical shifts were referenced to the residual solvent signal of DMSO at d 2.50 ppm for ^1^H and d 39.50 ppm for ^13^C. Individual resonances were assigned on the basis of their chemical shifts, signal intensities, multiplicity of resonances, and H_H coupling constants. High-performance liquid chromatography was performed on an Agilent 1100 series system with UV detection (photodiode array detector) using the Zorbax C18 reverse-phase analytical column (2.1_30 mm, 3.5 mm). All the compounds used for biological evaluation showed >95% purity in this HPLC system. The 4800 Plus MALDI TOF/TOF analyzer (Applied Biosystems Inc., Foster City, CA, USA), equipped with a 200 Hz, 355 nm Nd:YAG laser, was used for mass accuracy analysis of the compounds. Acquisition was performed in positive ion reflector mode. Elemental analyses were performed with the Perkin Elmer 2400 Series Organic Elemental analyzer.

4,7-dichloroquinoline **1**, benzene-1,2-diamine **4d**, 4-chlorobenzene-1,2-diamine **4e**, and 4-formylbenzoic acid **9** are commercially available; the synthesis of compounds **2**,**3** [30], **4a–c** [32,33], **6** [34], and **11** [35] was reported earlier. All the spectroscopic and analytical data of the above-mentioned compounds correspond with information previously reported and will not be shown in this paper. The procedures for the preparation of the remaining compounds illustrated in Figure 1 and Figure 2 are presented below.

#### 3.1.1. General Procedure for the Synthesis of Compounds **5a–e**

The solution of compound **3** (1 mmol) with the appropriate benzene-1,2-diamine **4a–e** (1 mmol) and Na_2_S_2_O_5_ (0.5 mmol) in DMSO (15 mL) is heated at 165 °C for 15 min. The mixture is then cooled to room temperature and water (5 mL) is added, resulting in the precipitation of a crude residue. The crude residues of compounds **5a–c** are collected by filtration, dissolved in methanol, and purified using column chromatography (CH_2_Cl_2_/MeOH 3:1; MeOH/NH_4_OH 3:1. The resulting products are converted to hydrochloride salts using anhydrous methanol saturated with HCl (g). The residues of compounds **5d** and **5e** are collected by filtration and purified by recrystallization using methanol.

##### 2-(3-(7-chloroquinolin-4-ylamino)phenyl)-1H-benzo[d]imidazole-5-carboximidamide trihydrochloride (**5a**)

Compound **5a** was prepared using the above-described method from **3** (190 mg, 0.67 mmol), diamine amidine **4a** (126 mg, 0.67 mmol), and Na_2_S_2_O_5_ (64 mg, 0.34 mmol) as a dark brown product (95 mg, 25%); mp > 250 °C. ^1^H NMR (300 MHz, DMSO) δ 13.80 (s, 1H, NH), 10.69 (s, 1H, NH), 9.35 (s, 2H, NH), 9.03 (s, 2H, NH), 8.78 (d, *J* = 9.1 Hz, 1H, ArH), 8.58 (d, *J* = 5.9 Hz, 1H, ArH), 8.37 (s, 1H, ArH), 8.23 (d, *J* = 7.3 Hz, 2H, ArH), 8.10 (s, 1H, ArH), 7.92–7.59 (m, 5H, ArH), 7.01 (d, *J* = 6.4 Hz, 1H, ArH). ^13^C NMR (150 MHz, DMSO) δ 166.44, 152.92, 146.98, 143.65, 139.43, 139.32, 137.57, 131.39, 131.19, 127.30, 126.89, 126.03, 125.06, 122.95, 122.62, 122.27, 121.75, 120.10, 117.36, 112.55, 101.73. Anal. calcd. for C_23_H_17_ClN_6_ × 3HCl × 2H_2_O (*M*_r_ = 558.29): C 49.48, H 4.33, N 15.05; found: C 49.31, H 4.41, N 15.22. HRMS: calcd. for C_23_H_18_ClN_6_ (M + H)^+^: 413.1281; found: 412.1283.

##### 2-(3-(7-chloroquinolin-4-ylamino)phenyl)-N-isopropyl-1H-benzo[d]imidazole-5-carboximidamide trihydrochloride (**5b**)

Compound **5b** was prepared using the above-described method from **3** (259 mg, 0.92 mmol), diamine amidine **4b** (210 mg, 0.92 mmol), and Na_2_S_2_O_5_ (87 mg, 0.46 mmol) as a light brown product (160 mg, 28%); mp > 250 °C. ^1^H NMR (300 MHz, DMSO-*d*_6_) δ/ppm 13.85 (d, *J* = 35.2 Hz, 1H, NH), 10.04 (s, 1H, NH), 9.48 (dd, *J* = 39.1, 13.2 Hz, 2H, NH), 9.01 (s, 1H, NH), 8.67 (d, *J* = 9.1 Hz, 1H, ArH), 8.56 (d, *J* = 5.8 Hz, 1H, ArH), 8.33 (s, 1H, ArH), 8.21–7.81 (m, 4H, ArH), 7.58 -7.73 (m, 4H, ArH), 7.04 (d, *J* = 5.9 Hz, 1H), 4.09 (m, 1H, CH), 1.31 (d, *J* = 6.3 Hz, 6H). ^13^C NMR (75 MHz, DMSO) δ 162.89, 154.39, 153.77, 150.25, 147.39, 143.52, 140.66, 138.88, 135.91, 135.04, 131.22, 130.95, 126.38, 125.57, 123.70, 123.07, 122.37, 121.73, 120.09, 119.45, 118.26, 112.84, 112.31, 102.37, 45.47, 21.78. Anal. calcd. for C_26_H_23_ClN_6_ × 3HCl × 3.5H_2_O (*M*_r_ = 627.39): C 49.77, H 5.30, N 13.40; found: C 49.69, H 5.05, N 13.22. HRMS: calcd. for C_26_H_24_ClN_6_ (M + H)^+^: 455.1751; found: 454.1750.

##### 7-chloro-N-(3-(5-(4,5-dihydro-1H-imidazol-2-yl)-1H-benzo[d]imidazol-2-yl)phenyl)quinolin-4-amine trihydrochloride (**5c**)

Compound **5c** was prepared using the above-described method from **3** (265 mg, 0.94 mmol), diamine amidine **4c** (200 mg, 0.94 mmol), and Na_2_S_2_O_5_ (0.47 mmol, 89 mg) as a brown product (140 mg, 26%); mp > 250 °C. ^1^H NMR (300 MHz, DMSO-*d*_6_) δ/ppm 14.78 (s, 1H, NH), 11.36 (s, 1H, NH), 10.65 (s, 2H, NH), 8.93 (d, *J* = 9.2 Hz, 1H, ArH), 8.60 (d, *J* = 7.0 Hz, 1H, ArH), 8.42 (d, *J* = 6.9 Hz, 2H, ArH), 8.35 (d, *J* = 7.9 Hz, 1H, ArH), 8.20 (d, *J* = 1.9 Hz, 1H, ArH), 7.94 (d, *J* = 9.0 Hz, 1H, ArH), 7.90–7.64 (m, 4H, ArH), 6.96 (d, *J* = 7.0 Hz, 1H, ArH), 4.03 (s, 4H, CH_2_). ^13^C NMR (75 MHz, DMSO-*d*_6_) δ/ppm 165.66, 155.28, 153.88, 144.07, 139.50, 139.00, 138.35, 131.37, 130.78, 128.36, 128.03, 126.73, 126.57, 124.47, 123.47, 119.72, 116.53, 116.48, 115.74, 101.15, 44.72.

Anal. calcd. for C_25_H_19_ClN_6_ × 3HCl × 1.5H_2_O (*M*_r_ = 575.32): C 52.19, H 4.38, N 14.61; found: C 51.97, H 4.25, N 14.22. HRMS: calcd. for C_25_H_20_ClN_6_ (M + H)^+^: 439.1438; found: 439.1437.

##### N-(3-(1H-benzo[d]imidazol-2-yl)phenyl)-7-chloroquinolin-4-amine (**5d**)

Compound **5d** was prepared using the above-described method from **3** (250 mg, 0.9 mmol), diamine **4d** (95 mg, 0.9 mmol), and Na_2_S_2_O_5_ (0.45 mmol, 85 mg) as a yellow product (130 mg, 36%); mp 248–250 °C. ^1^H NMR (300 MHz, DMSO-*d*_6_) δ/ppm 13.03 (s, 1H, NH), 10.17 (s, 1H, NH), 8.62 (d, *J* = 9.1 Hz, 1H, ArH), 8.57 (d, *J* = 6.2 Hz, 1H, ArH), 8.28 (s, 1H, ArH), 8.08 (d, *J* = 7.8 Hz, 1H, ArH), 8.00 (d, *J* = 2.1 Hz, 1H, ArH), 7.79 (dd, *J* = 9.1, 2.1 Hz, 1H, ArH), 7.74–7.52 (m, 4H, ArH), 7.22 (dd, *J* = 6.0, 3.1 Hz, 2H, ArH), 7.02 (d, *J* = 6.2 Hz, 1H, ArH). ^13^C NMR (151 MHz, DMSO-*d*_6_) δ/ppm 151.57, 151.24, 149.01, 148.81, 141.09, 135.00, 131.88, 130.70, 127.20, 125.90, 125.09, 124.49, 122.68, 120.97, 118.67, 102.64. Anal. calcd. for C_22_H_15_ClN_4_ × 1.5H_2_O (*M*_r_ = 397.86): C 66.41, H 4.56, N 14.08; found: C 66.21, H 4.33, N 14.19. HRMS: calcd. for C_22_H_16_ClN_4_ (M + H)^+^: 371.1063; found: 371.1056.

##### 7-chloro-N-(3-(5-chloro-1H-benzo[d]imidazol-2-yl)phenyl)quinolin-4-amine (**5e**)

Compound **5e** was prepared using the above-described method from **3** (296 mg, 1.05 mmol), diamine **4e** (150 mg, 1.05 mmol), and Na_2_S_2_O_5_ (0.525 mmol, 100 mg) as a yellow product (160 mg, 33%); mp 244–246 °C. ^1^H NMR (300 MHz, DMSO-*d*_6_) δ/ppm 13.20 (s, 1H, NH), 10.09 (s, 1H, NH), 8.64–8.54 (m, 2H, ArH), 8.26 (s, 1H, ArH), 8.06 (d, *J* = 7.9 Hz, 1H, ArH), 7.99 (d, *J* = 2.1 Hz, 1H, ArH), 7.77 (dd, *J* = 9.1, 2.1 Hz, 1H, ArH), 7.73–7.51 (m, 4H, ArH), 7.25 (d, *J* = 8.4 Hz, 1H, ArH), 7.03 (d, *J* = 6.1 Hz, 1H, ArH). ^13^C NMR (151 MHz, DMSO-*d*_6_) δ/ppm 152.44, 151.26, 149.00, 145.33, 140.09, 136.52, 131.60, 130.98, 126.75, 125.70, 125.50, 124.59, 123.90, 123.05, 122.03, 117.89, 102.12, 65.36. Anal. calcd. for C_22_H_14_Cl_2_N_4_ × 2.75H_2_O (*M*_r_ = 454.82): C 58.11, H 4.32, N 12.32; found: C 57.79, H 4.31, N 12.22. HRMS: calcd. for C_22_H_15_Cl_2_N_4_ (M + H)^+^: 405.0674; found: 404.0669.

#### 3.1.2. 4-(4-(7-chloroquinolin-4-yl)piperazin-1-yl)benzaldehyde (**7**)

A suspension of compound **6** (1.9 g, 7.71 mmol), 4-fluorbenzaldehyde (2.86 mg 23 mmol), and K_2_CO_3_ (3.17 mg, 23 mmol) in DMF (40 mL) was heated at 120 °C for 24 h. The mixture was cooled down to room temperature and the reaction suspension was diluted with dichloromethane (100 mL) and washed three times with water (100mL). The organic layer was dried on sodium sulfate, filtered, and concentrated on a rotary evaporator. The obtained yellow oil was diluted with methanol (2 mL), and water (5 mL) was added; the resulting yellow solid was filtered off (2 g, 78%); mp 182–183 °C. ^1^H NMR (600 MHz, DMSO-*d*_6_) δ/ppm 9.76 (s, 1H, CHO), 8.74 (d, *J* = 5.0 Hz, 1H, ArH), 8.13 (d, *J* = 9.0 Hz, 1H, ArH), 8.01 (d, *J* = 2.2 Hz, 1H, ArH), 7.77 (d, *J* = 8.9 Hz, 2H, ArH), 7.58 (dd, *J* = 9.0, 2.2 Hz, 1H), 7.14 (d, *J* = 8.9 Hz, 2H), 7.06 (d, *J* = 5.0 Hz, 1H, ArH), 3.72–3.64 (m, 4H, CH_2_), 3.36–3.32 (m, 4H, CH_2_). ^13^C NMR (151 MHz, DMSO-*d*_6_) δ/ppm 190.84, 156.42, 155.08, 152.70, 150.14, 134.14, 131.97, 128.60, 127.14, 126.53, 126.36, 121.79, 113.99, 110.01, 51.77, 46.89. HRMS: calcd. for C_20_H_19_ClN_3_O (M + H)^+^: 352.1216; found: 352.1220.

#### 3.1.3. General Procedure for the Synthesis of Compounds **8a–e**

A solution of compound **7** (1 mmol) with the appropriate benzene-1,2-diamine **4a–e** (1 mmol) and Na_2_S_2_O_5_ (0.5 mmol) in DMSO (15 mL) was heated at 165 °C for 15 min. The mixture was cooled down to room temperature. The addition of water (5 mL) resulted in precipitation. The resulting residue for compounds **8a–c** was collected with filtration and dissolved in methanol saturated with HCl; the addition of diethyl ether resulted in the precipitation of the products as hydrochloride salts. The residues of compounds **8d** and **8e** were collected with filtration, and the products were obtained by recrystallization using methanol.

##### 2-(4-(4-(7-chloroquinolin-4-yl)piperazin-1-yl)phenyl)-1H-benzo[d]imidazole-5-carboxamidine dihydrochloride (**8a**)

Compound **8a** was prepared using the above-described method from **7** (300 mg, 0.86 mmol), diamine amidine **4a** (160 mg, 0.86 mmol), and Na_2_S_2_O_5_ (82 mg, 0.43 mmol) as a light brown product (245 mg, 46%); mp > 250 °C. ^1^H NMR (600 MHz, DMSO-*d*_6_) δ/ppm 9.58 (s, 2H, NH), 9.27 (s, 2H, NH), 8.65 (d, *J* = 7.0 Hz, 1H, ArH), 8.41 (d, *J* = 8.8 Hz, 2H, ArH), 8.32 (d, *J* = 9.2 Hz, 1H, ArH), 8.22 (d, *J* = 2.1 Hz, 1H, ArH), 8.18 (s, 1H, ArH), 7.88–7.68 (m, 2H, ArH), 7.69 (dd, *J* = 9.2, 2.1 Hz, 1H, ArH), 7.15 (d, *J* = 7.1 Hz, 1H), 7.13 (d, *J* = 9.0 Hz, 2H), 4.12 (s, 4H, CH_2_), 3.81 (s, 4H, CH_2_). ^13^C NMR (151 MHz, DMSO-*d*_6_) δ/ppm 165.99, 159.94, 153.06, 152.61, 142.22, 140.68, 138.17, 130.51, 129.57, 126.56, 124.96, 124.32, 119.59, 117.50, 114.55, 114.17, 113.43, 105.13, 50.71, 45.32. Anal. calcd. for C_27_H_24_ClN_7_ × 2HCl × 4H_2_O (*M*_r_ = 626.96): C 51.72, H 5.47, N 15.64; found: C 51.68, H 5.49, N 15.32. HRMS: calcd. for C_27_H_25_ClN_7_ (M + H)^+^: 482.1860; found: 482.1860.

##### 2-(4-(4-(7-chloroquinolin-4-yl)piperazin-1-yl)phenyl)-N-isopropyl-1H-benzo[d]imidazole-5-carboxamidine dihydrochloride (**8b**)

Compound **8b** was prepared using the above-described method from **7** (350 mg, 0.99 mmol), diamine amidine **4b** (227 mg, 0.99 mmol), and Na_2_S_2_O_5_ (94 mg, 0.5 mmol) as an orange product (240 mg, 36%); mp 258 °C decomp. ^1^H NMR (600 MHz, DMSO-*d*_6_) δ/ppm 9.83 (d, *J* = 7.9 Hz, 1H, NH), 9.66 (s, 1H, NH), 9.28 (s, 1H, NH), 8.67 (d, *J* = 7.0 Hz, 1H, ArH), 8.52 (d, *J* = 8.7 Hz, 2H, ArH), 8.34 (d, *J* = 9.2 Hz, 1H, ArH), 8.29 (d, *J* = 2.1 Hz, 1H, ArH), 8.09 (d, *J* = 0.9 Hz, 1H, ArH), 7.89 (d, *J* = 8.5 Hz, 1H, ArH), 7.78 (dd, *J* = 8.5, 1.2 Hz, 1H, ArH), 7.72 (dd, *J* = 9.2, 2.1 Hz, 1H, ArH), 7.16 (dd, *J* = 10.8, 8.2 Hz, 3H, ArH), 4.15 (s, 5H, CH and CH_2_), 3.84 (s, 4H, CH_2_), 1.32 (d, *J* = 6.4 Hz, 6H, CH_3_). ^13^C NMR (151 MHz, DMSO-*d*_6_) δ/ppm 162.05, 159.92, 153.17, 152.03, 142.13, 140.68, 138.12, 135.85, 132.45, 130.72, 129.56, 126.55, 125.88, 125.48, 119.56, 117.50, 114.47, 113.91, 113.36, 110.83, 105.11, 50.67, 45.72, 45.26, 21.73. Anal. calcd. for C_30_H_30_ClN_7_ × 2HCl × 3.5H_2_O (*M*_r_ = 660.03): C 54.59, H 5.96, N 14.85; found: C 54.68, H 5.83, N 14.72. HRMS: calcd. for C_30_H_31_ClN_7_ (M + H)^+^: 524.2329; found: 524.2326.

##### 7-chloro-4-(4-(4-(5-(4,5-dihydro-1H-imidazol-2-yl)-1H-benzo[d]imidazol-2-yl)phenyl)piperazin-1-yl)quinoline dihydrochloride (**8c**)

Compound **8c** was prepared using the above-described method from **7** (280 mg, 0.8 mmol), diamine amidine **4c** (170 mg, 0.8 mmol), and Na_2_S_2_O_5_ (76 mg, 0.4 mmol) as an orange product (230 mg, 45%); mp 250 °C decomp. ^1^H NMR (300 MHz, DMSO-*d*_6_) δ/ppm 10.72 (s, 2H, NH), 8.70 (d, *J* = 6.9 Hz, 1H, ArH), 8.37–8.18 (m, 5H, ArH), 7.90 (d, *J* = 8.4 Hz, 1H, ArH), 7.80 (d, *J* = 8.4 Hz, 1H, ArH), 7.72 (dd, *J* = 9.2, 1.9 Hz, 1H, ArH), 7.20 (d, *J* = 7.0 Hz, 1H, ArH), 7.12 (d, *J* = 8.9 Hz, 2H, ArH), 4.08 (s, 4H, CH_2_), 4.03 (s, 4H, CH_2_), 3.74 (s, 4H, CH_2_). ^13^C NMR (151 MHz, DMSO-*d*_6_) δ/ppm 165.72, 159.87, 152.15, 143.09, 137.94, 129.30, 126.63, 123.25, 120.30, 117.93, 113.97, 105.73, 51.06, 46.05, 44.72. Anal. calcd. for C_29_H_26_ClN_7_ × 2HCl × 3H_2_O (*M*_r_ = 634.98): C 54.85, H 5.40, N 15.44; found: C 54.76, H 5.38, N 15.22. HRMS: calcd. for C_29_H_27_ClN_7_ (M + H)^+^: 508.2016; found: 508.2008.

##### 4-(4-(4-(1H-benzo[d]imidazol-2-yl)phenyl)piperazin-1-yl)-7-chloroquinoline (**8d**)

Compound 8d was prepared using the above-described method from 7 (400 mg, 1.14 mmol), diamine 4d (123 mg, 1.14 mmol), and Na_2_S_2_O_5_ (108 mg, 0.57 mmol) as an orange product (310 mg, 57%); mp 254 °C decomp.^1^H NMR (300 MHz, DMSO-d_6_) δ/ppm 8.74 (d, *J* = 5.2 Hz, 1H, ArH), 8.15 (d, *J* = 9.0 Hz, 1H, ArH), 8.08 (d, *J* = 8.8 Hz, 2H, ArH), 8.02 (d, *J* = 2.1 Hz, 1H, ArH), 7.62–7.55 (m, 3H, ArH), 7.21–7.18 (m, 4H, ArH), 7.10 (d, *J* = 5.2 Hz, 1H, ArH), 3.60 (s, 4H, CH_2_), 3.41 (s, 4H, CH_2_). ^13^C NMR (151 MHz, DMSO-d_6_) δ/ppm 156.84, 152.25, 151.98, 151.90, 149.36, 139.13, 134.48, 128.23, 127.89, 126.75, 126.39, 122.44, 121.50, 119.79, 115.20, 114.91, 109.67, 51.89, 47.64. Anal. calcd. for C_26_H_22_ClN_5_ × 2.25H_2_O (M_r_ = 480.47): C 64.99, H 5.56, N 14.58; found: C 65.11, H 5.39, N 14.21. HRMS: calcd. for C_26_H_23_ClN_5_ (M + H)^+^: 440.1642; found: 440.1637.

##### 7-chloro-4-(4-(4-(5-chloro-1H-benzo[d]imidazol-2-yl)phenyl)piperazin-1-yl)quinoline (**8e**)

Compound **8e** was prepared using the above-described method from **7** (400 mg, 1.14 mmol), diamine **4e** (162 mg, 1.14 mmol), and Na_2_S_2_O_5_ (108 mg, 0.57 mmol) as a brown product (290 mg, 48%); mp 206–208 °C.^1^H NMR (300 MHz, DMSO-*d*_6_) δ/ppm 8.74 (d, *J* = 5.3 Hz, 1H, ArH), 8.16 (d, *J* = 9.0 Hz, 1H, ArH), 8.06 (d, *J* = 8.9 Hz, 2H, ArH), 8.02 (d, *J* = 2.2 Hz, 1H, ArH), 7.61 (dd, *J* = 9.0, 2.2 Hz, 1H, ArH), 7.57 (d, *J* = 1.9 Hz, 1H, ArH), 7.53 (d, *J* = 8.5 Hz, 1H, ArH), 7.18 (dd, *J* = 8.1, 1.5 Hz, 3H, ArH), 7.11 (d, *J* = 5.3 Hz, 1H, ArH), 3.61 (s, 4H, CH_2_), 3.45 (s, 5H, CH_2_). ^13^C NMR (151 MHz, DMSO-*d*_6_) δ/ppm 157.29, 153.60, 152.20, 150.69, 148.22, 134.99, 128.27, 127.12, 126.80, 126.43, 126.41, 122.29, 120.99, 119.79, 115.04, 109.13, 56.49, 51.79, 47.43, 19.01. Anal. calcd. for C_26_H_21_Cl_2_N_5_ × 3H_2_O (*M*_r_ = 528.43): C 59.1, H 5.15, N 13.25; found: C 58.78, H 4.96, N 13.12. HRMS: calcd. for C_26_H_22_Cl_2_N_5_ (M + H)^+^: 474.1252; found: 474.1250

#### 3.1.4. General Procedure for the Synthesis of Compounds **10a–d**

A solution of compound **9** (1 mmol) with the appropriate benzene-1,2-diamine **4a–d** (1 mmol) and Na_2_S_2_O_5_ (0.5 mmol) in DMSO (15 mL) was heated at 165 °C for 15 min. The mixture was cooled down to room temperature. The addition of water (5 mL) resulted in precipitation. The resulting residue for compounds **10a-c** was collected with filtration and dissolved in methanol saturated with HCl; the addition of diethyl ether resulted in the precipitation of the products as hydrochloride salts. The residue of compound **8d** was collected with filtration, and the product was obtained by recrystallization using methanol.

##### 4-(5-carbamimidoyl-1H-benzo[d]imidazol-2-yl)benzoic acid dihydrochloride (**10a**)

Compound **10a** was prepared using the above-described method from **9** (300 mg, 2 mmol), diamine amidine **4a** (373 mg, 2 mmol), and Na_2_S_2_O_5_ (190 mg, 1 mmol) as a light brown product (445 mg, 60%); mp > 250 °C. ^1^H NMR (600 MHz, DMSO-*d*_6_) δ/ppm 9.50 (s, 2H, NH), 9.24 (s, 2H, NH), 8.48 (d, *J* = 8.4 Hz, 2H, ArH), 8.27 (d, *J* = 1.3 Hz, 1H, ArH), 8.15 (d, *J* = 8.5 Hz, 2H, ArH), 7.89 (d, *J* = 8.5 Hz, 1H, ArH), 7.81 (d, *J* = 1.7 Hz, 1H, ArH). ^13^C NMR (151 MHz, DMSO-*d*_6_) δ/ppm 166.56, 165.88, 152.38, 133.06, 131.24, 129.95, 127.59, 123.25, 122.68, 116.3, 114.93. HRMS: calcd. for C_15_H_13_N_4_O_2_ (M + H)^+^: 281.1038; found: 281.1031.

##### 4-(5-(N-isopropylcarbamimidoyl)-1H-benzo[d]imidazol-2-yl)benzoic acid dihydrochloride (**10b**)

Compound **10b** was prepared using the above-described method from **9** (300 mg, 2 mmol), diamine amidine **4b** (457 mg, 2 mmol), and Na_2_S_2_O_5_ (190 mg, 1 mmol) as a grey product (520 mg, 63%); mp > 250 °C. ^1^H NMR (300 MHz, DMSO-*d*_6_) δ/ppm 9.74 (d, *J* = 8.0 Hz, 1H, NH), 9.60 (s, 1H, NH), 9.24 (s, 1H, NH), 8.56 (d, *J* = 8.4 Hz, 2H, ArH), 8.18 (d, *J* = 8.6 Hz, 3H, ArH), 7.92 (d, *J* = 8.5 Hz, 1H, ArH), 7.73 (dd, *J* = 8.6, 1.5 Hz, 1H, ArH), 4.22–4.09 (m, 1H, CH), 1.32 (d, *J* = 6.4 Hz, 6H, CH_3_). ^13^C NMR (75 MHz, DMSO-*d*_6_) δ/ppm 166.98, 162.33, 152.04, 139.08, 136.27, 133.96, 130.64, 130.47, 128.41, 125.20, 124.66, 116.46, 115.21, 45.65, 21.76. HRMS: calcd. for C_18_H_19_N_4_O_2_ (M + H)^+^: 323.1508; found: 322.1515.

##### 4-(5-(4,5-dihydro-1H-imidazol-2-yl)-1H-benzo[d]imidazol-2-yl)benzoic acid hydrochloride (**10c**)

Compound **10c** was prepared using the above-described method from **9** (300 mg, 2 mmol), diamine amidine **4c** (425 mg, 2 mmol), and Na_2_S_2_O_5_ (190 mg, 1 mmol) as a grey product (515 mg, 67%); mp > 250 °C.^1^H NMR (300 MHz, DMSO-*d*_6_) δ/ppm 10.88 (s, 2H, NH), 8.52–8.40 (m, 3H, ArH), 8.14 (d, *J* = 8.4 Hz, 2H, ArH), 8.00 (dd, *J* = 8.6, 1.3 Hz, 1H, ArH), 7.89 (d, *J* = 8.5 Hz, 1H, ArH), 4.03 (s, 4H, CH_2_). ^13^C NMR (75 MHz, DMSO-*d*_6_) δ/ppm 167.08, 165.41, 153.28, 141.44, 138.29, 133.45, 131.85, 130.40, 128.07, 124.06, 117.56, 117.09, 115.73, 44.73. HRMS: calcd. for C_17_H_15_N_4_O_2_ (M + H)^+^: 307.1195; found: 307.1206.

##### 4-(1H-benzo[d]imidazol-2-yl)benzoic acid (**10d**)

Compound **10d** was prepared using the above-described method from **9** (250 mg, 1.66 mmol), diamine **4d** (180 mg, 1.66 mmol), and Na_2_S_2_O_5_ (157 mg, 0.83 mmol) as a grey product (350 mg, 82%); mp 215–217 °C. ^1^H NMR (600 MHz, DMSO-*d*_6_) δ/ppm 8.31 (d, *J* = 8.5 Hz, 2H, ArH), 8.15 (d, *J* = 8.5 Hz, 2H, ArH), 7.70 (dd, *J* = 6.0, 3.2 Hz, 2H, ArH), 7.33 (dd, *J* = 6.1, 3.1 Hz, 2H, ArH). ^13^C NMR (151 MHz, DMSO-*d*_6_) δ/ppm δ 167.20, 150.12, 138.17, 132.73, 130.48, 127.35, 127.35, 123.88, 115.59. HRMS: calcd. for C_14_H_11_N_2_O_2_ (M + H)^+^: 239.0820; found: 239.0824.

#### 3.1.5. General Procedure for the Synthesis of Compounds **12a–d**

To a suspension of the appropriate benzoic acid **10a–d** (1 mmol) in dichloromethane (30 mL), triethylamine (Et_3_N) (6.75 mmol), and 1-hydroxybenzotriazole (HOBt) (1.1 mmol), compound 11 (1 mmol) and *N*-(3-Dimethylaminopropyl)-*N*′-ethylcarbodiimide hydrochloride (EDCxHCl) (1.4 mmol) were added. The reaction mixture was stirred at room temperature for 48 h. An ammonium hydroxide solution (pH 8.0) (30 mL) was added to reaction mixture. The resulting sticky residue was separated with decantation. The residue for compounds **12a–c** was dissolved in methanol saturated with HCl; the addition of diethyl ether resulted in the precipitation of the products as hydrochloride salts. The residue of compound **12d** was obtained by recrystallization using methanol.

##### 4-(5-carbamimidoyl-1H-benzo[d]imidazol-2-yl)-N-(2-(7-chloroquinolin-4-ylamino)ethyl)benzamide trihydrochloride (**12a**)

Compound **12a** was prepared using the above-described method from **10a** (371 mg, 1 mmol), Et_3_N (688 mg, 6.75 mmol), HOBt (149 mg, 1.1 mmol), compound **11** (222 mg, 1 mmol), and EDCxHCl (267 mg, 1.4 mmol) in dichloromethane (30 mL) as a grey product (290 mg, 45%); mp > 250 °C ^1^H NMR (600 MHz, DMSO-*d*_6_) δ/ppm 14.33 (s, 1H, NH), 9.81 (t, *J* = 5.7 Hz, 1H, NH), 9.41 (s, 2H, NH), 9.15 (t, *J* = 5.6 Hz, 1H, NH), 9.12 (d, *J* = 18.2 Hz, 2H, NH), 8.72 (d, *J* = 9.2 Hz, 1H, ArH), 8.56 (dd, *J* = 11.9, 6.7 Hz, 1H, ArH), 8.39 (d, *J* = 8.4 Hz, 2H, ArH), 8.21 (d, *J* = 1.2 Hz, 1H, ArH), 8.07 (dd, *J* = 9.4, 5.3 Hz, 3H, ArH), 7.84 (d, *J* = 8.5 Hz, 1H, ArH), 7.76 (dd, *J* = 9.0, 2.1 Hz, 1H, ArH), 7.73 (dd, *J* = 8.5, 1.7 Hz, 1H, ArH), 7.00 (d, *J* = 7.2 Hz, 1H, ArH), 3.76 (dd, *J* = 12.0, 6.0 Hz, 2H, CH_2_), 3.63 (dd, *J* = 11.9, 6.0 Hz, 2H, CH_2_). ^13^C NMR (151 MHz, DMSO-*d*_6_) δ/ppm 166.43, 166.38, 156.26, 153.35, 143.25, 138.97, 138.45, 136.57, 131.10, 128.51, 127.66, 127.33, 126.47, 124.24, 123.34, 122.73, 119.45, 116.06, 99.07, 56.47, 43.07, 38.31, 19.01. Anal. calcd. for C_26_H_22_ClN_7_O × 3HCl × 3H_2_O (*M*_r_ = 647.38): C 48.24, H 4.83, N 15.15; found: C 48.01, H 4.47, N 15.23. HRMS: calcd. for C_26_H_23_ClN_7_O (M + H)^+^: 484.1653; found: 484.1640.

##### 4-(5-(N-isopropylcarbamimidoyl)-1H-benzo[d]imidazol-2-yl)-N-(2-(7-chloroquinolin-4-ylamino)ethyl)benzamide trihydrochloride (**12b**)

Compound **12b** was prepared using the above-described method from **10b** (413 mg, 1 mmol), Et_3_N (688 mg, 6.75 mmol), HOBt (149 mg, 1.1 mmol), compound **11** (222 mg, 1 mmol), and EDCxHCl (267 mg, 1.4 mmol) in dichloromethane (30 mL) as a light brown product (266 mg, 39%); mp > 250 °C. ^1^H NMR (600 MHz, DMSO-*d*_6_) δ/ppm 14.50 (s, 1H, NH), 9.90 (t, *J* = 5.7 Hz, 1H, NH), 9.68 (d, *J* = 8.1 Hz, 1H, NH), 9.54 (s, 1H, NH), 9.26 (t, *J* = 5.6 Hz, 1H, NH), 9.15 (s, 1H, NH), 8.79 (d, *J* = 9.2 Hz, 1H, ArH), 8.56 (s, 1H, ArH), 8.48 (d, *J* = 8.4 Hz, 2H, ArH), 8.15–8.12 (m, 3H, ArH), 8.11 (d, *J* = 2.1 Hz, 1H, ArH), 7.88 (d, *J* = 8.5 Hz, 1H, ArH), 7.75 (dd, *J* = 9.1, 2.1 Hz, 1H, ArH), 7.69 (dd, *J* = 8.5, 1.6 Hz, 1H, ArH), 7.01 (d, *J* = 7.2 Hz, 1H, ArH), 4.13 (m, 1H, CH), 3.77 (dd, *J* = 11.8, 5.9 Hz, 2H, CH_2_)., 1.31 (d, *J* = 6.4 Hz, 6H, CH_3_). ^13^C NMR (151 MHz, DMSO-*d*_6_) δ/ppm 166.27, 162.46, 156.22, 152.47, 143.16, 138.97, 138.39, 137.03, 129.73, 128.57, 128.02, 127.27, 126.57, 124.84, 124.29, 119.40, 116.47, 116.06, 115.20, 99.05, 45.62, 43.03, 38.32, 21.76. Anal. calcd. for C_29_H_28_ClN_7_O × 3HCl × 1.75H_2_O (*M*_r_ = 666.94): C 52.23, H 5.21, N 14.70; found: C 52.02, H 5.57, N 14.53. HRMS: calcd. for C_29_H_29_ClN_7_O (M + H)^+^: 526.2122; found: 526.2122.

##### N-(2-(7-chloroquinolin-4-ylamino)ethyl)-4-(5-(4,5-dihydro-1H-imidazol-2-yl)-1H-benzo[d]imidazol-2-yl)benzamide trihydrochloride (**12c**)

Compound **12c** was prepared using the above-described method from **10c** (383 mg, 1 mmol), Et_3_N (688 mg, 6.75 mmol), HOBt (149 mg, 1.1 mmol), compound **11** (222 mg, 1 mmol), and EDCxHCl (267 mg, 1.4 mmol) in dichloromethane (30 mL) as a brown product (304 mg, 44%); mp > 250 °C.^1^H NMR (600 MHz, DMSO-*d*_6_) δ/ppm 14.40 (s, 1H, NH), 10.76 (s, 2H, NH), 9.84 (t, *J* = 5.7 Hz, 1H, NH), 9.17 (t, *J* = 5.6 Hz, 1H, NH), 8.74 (d, *J* = 9.2 Hz, 1H, ArH), 8.60–8.52 (m, 1H, ArH), 8.45 (d, *J* = 1.0 Hz, 1H, ArH), 8.40 (d, *J* = 8.4 Hz, 2H, ArH), 8.10–8.06 (m, 3H, ArH), 7.93 (dd, *J* = 8.5, 1.4 Hz, 1H, ArH), 7.86 (d, *J* = 8.5 Hz, 1H, ArH), 7.75 (dd, *J* = 9.1, 2.1 Hz, 1H, ArH), 7.00 (d, *J* = 7.2 Hz, 1H, ArH), 4.01 (s, 4H, CH_2_), 3.75 (dd, *J* = 11.9, 6.0 Hz, 2H, CH_2_), 3.63 (dd, *J* = 11.8, 5.9 Hz, 2H, CH_2_). ^13^C NMR (151 MHz, DMSO-*d*_6_) δ/ppm 166.40, 165.62, 156.24, 153.74, 143.22, 138.97, 138.42, 136.61, 131.02, 128.50, 127.74, 127.30, 126.50, 123.72, 119.44, 116.77, 116.06, 99.06, 44.75, 43.06, 38.30. Anal. calcd. for C_28_H_24_ClN_7_O × 3HCl × 4H_2_O (*M*_r_ = 691.43): C 48.64, H 5.10, N 14.18; found: C 48.33, H 5.26, N 14.04. HRMS: calcd. for C_28_H_25_ClN_7_O (M + H)^+^: 510.1809; found: 510.1807.

##### 4-(1H-benzo[d]imidazol-2-yl)-N-(2-(7-chloroquinolin-4-ylamino)ethyl)benzamide (**12d**)

Compound **12d** was prepared using the above-described method from **10d** (256 mg, 1 mmol), Et_3_N (688 mg, 6.75 mmol), HOBt (149 mg, 1.1 mmol), compound **11** (222 mg, 1 mmol), and EDCxHCl (267 mg, 1.4 mmol) in dichloromethane (30 mL) as a light yellow product (245 mg, 51%); mp 195 °C decomp. ^1^H NMR (600 MHz, DMSO-*d*_6_) δ/ppm 13.02 (s, 1H, NH), 8.83 (t, *J* = 5.4 Hz, 1H, NH), 8.45 (d, *J* = 5.8 Hz, 1H, NH), 8.28 (d, *J* = 9.1 Hz, 1H, ArH), 8.25 (dd, *J* = 6.0, 4.2 Hz, 2H, ArH), 8.00(m, 2H, ArH), 7.93 (s, 1H, ArH), 7.81 (d, *J* = 2.2 Hz, 1H, ArH), 7.67 (s, 1H, ArH), 7.53 (dd, *J* = 9.0, 2.2 Hz, 2H, ArH), 7.21 (s, 2H, ArH), 6.72 (d, *J* = 5.8 Hz, 1H, ArH), 3.66–3.44 (m, 4H, CH_2_). ^13^C NMR (151 MHz, DMSO-*d*_6_) δ/ppm 166.21, 151.27, 150.30, 150.01, 146.83, 143.75, 142.06, 135.04, 134.37, 132.57, 127.83, 126.23, 125.73, 124.73, 124.29, 122.69, 121.99, 118.99, 117.05, 111.45, 98.63, 42.10, 37.91. Anal. calcd. for C_25_H_20_ClN_5_O × 2H_2_O (*M*_r_ = 477.94): C 62.82, H 5.06, N 14.65; found: C 62.49, H 5.27, N 14.33. HRMS: calcd. for C_25_H_21_ClN_5_O (M + H)^+^: 442.1435; found: 442.1426.

### 3.2. Biological Activity

#### 3.2.1. Evaluation of the Antiproliferative Effect

The growth inhibition activity was assessed according to the slightly modified procedure performed at the National Cancer Institute, Developmental Therapeutics Program [67].

##### Cell Lines

The examined compounds were dissolved in DMSO (1 × 10^−2^ M). The experiments were carried out on seven human tumor cell lines and a normal cell line. The following cell lines were used: HeLa (human cervical adenocarcinoma; purchased from ATCC), CaCo-2 (human colorectal adenocarcinoma), MCF-7 (human breast adenocarcinoma), CCRF-CEM (human acute lymphoblastic *leukemia*), Hut78 (T-cell lymphoma), THP-1 (acute monocytic leukemia), Burkitt lymphoma (Raji), and MDCK1 (Madine–Darby canine kidney). The MDCK1 cells were used between 24 and 26 passages.

##### Cell Culturing

The adherent cells were cultured in Dulbecco’s modified Eagle medium—DMEM (Gibco, Termo Fisher Scientific Inc., Leicestershire, England, UK), supplemented with 10% heat-inactivated fetal bovine serum (FBS, Gibco, Termo Fisher Scientific Inc., Leicestershire, England, UK), 2 mM glutamine, and 100U/0.1mg penicillin/streptomycin. The cells in suspension were cultured in RPMI 1640 (Gibco, Termo Fisher Scientific Inc., Leicestershire, England, UK) medium, supplemented with 10% FBS (Gibco, Termo Fisher Scientific Inc., Leicestershire, England, UK), 2 mM glutamine, 1 mM sodium pyruvate, and 10 mM HEPES. The cells were grown in a humidified atmosphere under the conditions of 37 °C/5% of CO_2_ gas in the CO_2_ incubator (IGO 150 CELLlife^TM^, JOUAN, Thermo Fisher Scientific, Waltham, MA, USA).

##### Proliferation Assay

The adherent cells (HeLa, CaCo-2, MCF-7, and MDCK-1) were plated in 96-well flat bottom plates (Greiner, Frickenhausen, Austria) at a concentration of 2 × 10^4^ cells/mL. The suspension cells (Raji, THP-1, CCRF-CEM and Hut78) were plated in 96-well microtiter plates (Sarstead, Newton, NC, USA) at a concentration of 1 × 10^5^ cells/mL. Twenty-four hours later, the cells were treated with test agents in five 10-fold dilutions (10^−7^ to 10^−4^ M) and incubated for a further 72 h. The working dilutions were freshly prepared on the day of testing. The solvent was also tested for eventual inhibitory activity by adjusting its concentration to be the same as in the working concentrations. After 72 h of incubation, the cell growth rate was evaluated by performing the MTT assay, which detects dehydrogenase activity in viable cells [68]. For this purpose, upon completion of the incubation period, the growth medium was discarded and 50 μL of MTT was added to each well at a concentration of 5 mg/mL. After four hours of incubation at 37 °C, the water insoluble MTT-formazan crystals were dissolved in 150 μL of dimethyl-sulfoxide (DMSO) for the adherent cells and in 10% SDS with 0.01 mol/L HCl for the cells grown in suspension. The absorbance (OD, optical density) was measured on a microplate reader (iMark, BIO RAD, Hercules, CA, USA) at 595 nm.

The percentage of live cells was calculated as follows: % = OD (sample) − OD (background)/OD (control) − OD (background) × 100.

The optical density (OD) of the background for the adherent cells is the OD of the MTT solution and DMSO; the OD (background) for the suspension cells is the OD of the culture medium with MTT and 10% SDS with 0.01 mol/L HCl; the OD (control) is the OD of the cells grown without the tested compounds.

The results were expressed as GI_50_, a concentration necessary for 50% inhibition. The calculation of the GI_50_ value curves and QC analysis was performed using the Excel tools and GraphPadPrism software (La Jolla, CA, USA), v. 5.03. Briefly, individual concentration effect curves are generated by plotting the logarithm of the concentration of the tested compounds (X) vs. the corresponding percent inhibition values (Y) using the least squares fit. The best-fit GI_50_ values are calculated using log (inhibitor) versus normalized response—variable slope equation, where Y Ľ 100/(1 ţ 10 ((LogIC_50_ _ X) × HillSlope)). The QC criteria parameters (Z0, S:B, R2, HillSlope) were checked for every GI_50_ curve.

#### 3.2.2. Flow Cytometry Analysis of Cell Cycle

To analyze the effects of the selected compounds (**5d**, **8d**, and **12d**) on the cell cycle of the HuT78 and THP-1 cells, the cells were seeded in 6-well plates (5 × 10^5^ cells per well) and treated for 24 h with the selected compounds **5d** and **12d** (5 μM). The cells were then collected and fixed in cold 70% ethanol. The fixed samples were stained with 15 µg/mL of propidium iodide (PI) at room temperature and analyzed by flow cytometry (FACS Canto II, BD Biosciences). The results were analyzed using FlowLogic software.

One-way ANOVA analysis of variance was performed in the MedCalc statistical program for the analysis of the cell cycle [69]. *p*-values of less than 0.05 were considered to be statistically significant.

#### 3.2.3. Measurement of Mitochondrial Membrane Potential (∆Ψm)

Changes in the (∆Ψm) were measured using TMRE (tetramethylrhodamine, ethyl ester and perchlorate) dye. In brief, the tested cells (HuT 78) were plated in 6-well plates at a concentration of 5 × 10^5^ cells per well and treated with 5 μM of the compounds **5d** and **12d**. After 24 h of treatment, the cells were collected, centrifuged for 6 min at 1100 rpm, and stained with 200 nM of TMRE dye according to the kit protocol (TMRE Mitochondrial Membrane Potential Assay Kit, abcam, Cambridge, UK). The positive control cells were treated with 20 μM FCCP (carbonyl cyanide-p-trifluoromethoxyphenylhydrazone) for 10 min. The cells were analyzed by flow cytometry (BD FACSCalibur, Becton Dickinson, San Jose, CA, USA) and FlowJo software (FlowJo, LLC, Ashland, OR, USA).

#### 3.2.4. Determination of Apoptosis

The proapoptotic potential of compounds **5d** and **12d** was tested on the HuT78 cells using the Annexin V-FITC Apoptosis Staining/Detection Kit (ab14085; abcam, UK). The cells were plated in 6-well plates at a concentration of 5 × 10^5^ cells/well and treated for 24 h with **12d** at a concentration of 5 μM. After the incubation period, the cells were collected and centrifuged at 1100 rpm for 6 min, stained according to the manufacturer‘s protocol, and analyzed by flow cytometry (BD FACSCalibur, Becton Dickinson, San Jose, CA, USA) using FlowJo software (FlowJo, LLC, Ashland, OR, USA).

### 3.3. Assessment of Absorption, Distribution, Metabolism, and Excretion (ADME) Properties

After the input molecules as SMILES, the structural, physicochemical, pharmacokinetics, and drug-likeness properties were calculated with the SwissADME web tool (http://www.swissadme.ch, accessed on 21 July 2020). Six physicochemical properties were taken into account for bioavailability: lipophilicity, size, polarity, solubility, flexibility, and saturation. SwissADME also provides water solubility in mol/L and mg/mL. “Drug-likeness” assesses qualitatively the chance for a molecule to become an oral drug with respect to bioavailability. Thus, the molecular weight (MW) should be between 150 and 500 g/mol; polarity and total polar surface area (TPSA) between 20 and 130 Å^2^; solubility in water and log S, not lower than −6; saturation: the ratio of sp^3^ hybridized carbons over the total carbon count of the molecule (not less than 0.25, and flexibility, no more than 9 rotatable bonds [70]. In this paper, we presented the drug likeness results according to the Lipinski (Pfizer) rule-based filter. The rule of 5′ predicts that there is good absorption or permeation if a molecule has: MW ≤ 500; H-bond donors ≤ 5; H-bond acceptors ≤ 10; and MLOGP ≤ 4.15 [55].

### 3.4. Molecular Docking Study

The molecular docking of the compounds (**5a–e**, **8a–e**, and **12a–d**), including the DSA compound (PDB ID: G6H) as a standalone ligand, was performed with *i* GEMDOCK (BioXGEM, Taiwan). The crystal coordinates of the c-Src (PDB ID: 3G6H) in the complex with the DSA compound were downloaded from the Protein Data Bank (PDB, https://www.rcsb.org/, accessed on 21 July 2020). In the first step, the structure of the c-Src was prepared, including the removal of the water molecules, and the protein structure was optimized using BIOVIA Discovery Studio 2017 R2 v(Dassault Systèmes, France). The binding site of the protein was defined according to the bonded ligand (PDB ID: G6H). After preparing the protein target and the setting of the optimized structures of 14 compounds as ligands, the genetic parameters were set (population size: 200; generations: 70; number of poses: 3). Each compound in the library was docked into the binding site and the protein–compound interaction profiles of the electrostatic (Elec), hydrogen-bonding (H-bond), and van der Waals (vdW) interactions were generated. Finally, the compounds were ranked by combining the pharmacological interactions and the energy-based scoring function. The energy-based scoring function or total energy (*E*/kcal mol^−1^) is: *E* = vdW + H-bond + Elec. [71].

## 4. Conclusions

Target amidine- and non-amidine-substituted 7-chloro-4-aminoquinoline-benzimidazole hybrids **5a–e**, **8a–e**, and **12a–d** were designed and prepared using various synthetic methods. The compounds **5a–e** and **8a–e** were synthesized by the oxidative coupling of o-phenylenediamines with 7-chloro-4-minoquinoline benzaldehydes **3** and **7**, and the compounds **12a–d** were prepared by the amidation of benzimidazole benzoic acids **10a–d** with 7-chloroquinoline diamine **11**.

The results of the in vitro studies of the antiproliferative activity toward one normal and seven cancer cell lines suggest that the amidine substituents on the benzimidazole ring decrease the activity of the novel compounds against normal and tumor cells. It should be highlighted that the compounds with unsubstituted benzimidazole rings (**5d**, **8d**, and **12d**) showed the highest antiproliferative activity against leukemia and lymphoma cell lines at micromolar or submicromolar concentrations. The potent antiproliferative activities of **5d** and **12d** are associated with statistically significant redistribution in the cell cycles and with the potential disruption of the mitochondrial membrane, accompanied by apoptosis induction. The ADME properties of compound **12d** suggest a good potential as an oral drug with respect to bioavailability. The molecular docking study indicates that an amino group from the *N*-(2-aminoethyl)benzamide central linker of compound **12d** is important for major interactions with the binding site of tyrosine-protein kinase c-Src.

## Data Availability

Not applicable.

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
