# Peer review of "Novel 7-Chloro-4-aminoquinoline-benzimidazole Hybrids as Inhibitors of Cancer Cells Growth: Synthesis, Antiproliferative Activity, in Silico ADME Predictions, and Docking"

_molecules, 2023, doi:10.3390/molecules28020540_

Round 1

Reviewer 1 Report

In this study, new 7-chloro-4-aminoquinoline-benzimidazole compounds have been synthesized and characterized by NMR, MS and elemental analysis.  Their antiproliferative activities were evaluated on one non-tumor (MDCK1) and seven selected tumor (CaCo-2, MCF-7, CCRF- 22 CEM, Hut78, THP-1, and Raji) cell lines by MTT test and flow cytometry analysis. The obtained results suggest that molecule 12d is the leading compound in the further design of effective antitumor drugs.

I have the following comments:

This paper needs considerable language check for grammars and punctuations. It was hard to follow in some places.

The authors did not use a positive control for the MTT assay. This control is very important to show how effective their compounds are in comparison to a standard chemotherapy.

What was the % of DMSO in all biological assays?

SD appears high to me in the IC50 results?

More discussion is needed on the cell cycle arrest results

Author Response

Dear Reviewer 1,

We would like to thank you for your valuable comments and suggestions that improved our manuscript.

Our responses to your comments and questions:

  1. This paper needs considerable language check for grammars and punctuations. It was hard to follow in some places.

Answer: A check of English grammar and punctuation was made. Done is marked in the manuscript with track changes.

  1. The authors did not use a positive control for the MTT assay. This control is very important to show how effective their compounds are in comparison to a standard chemotherapy.

Answer: As a well-known chemotherapeutic agent, 5-fluorouracil (5-FU) was used as s positive control and tested under the same experimental conditions as newly synthesized compounds. Obtained data are included in the Table 1 (Page 7).

  1. What was the % of DMSO in all biological assays?

Answer: Only stock solution of compounds (10 mM) were prepared in DMSO.  Working concentrations of compounds for biological experiments were prepared in ultrapure water. In all biological assays % of DMSO was between 0.1 and 0.01%.

  1. SD appears high to me in the IC50 results?

Answer: You have right for some obtained data. Here we presented the data obtained from 3 independent experiments.

  1. More discussion is needed on the cell cycle arrest results

Answer: The sub chapter 2.2.2. Cell cycle redistribution has been expanded (Page 8)

Note: we have added examples of recent research on the efforts “to introduce new anti-cancer entities”(Mao et al. 2022, Labozzeta et al. 2022, Moreno et al. 2022), we have also added one reference for the part that is talking about hybrid drugs (Szumilak et al. 2021). We have removed three references from the introduction part, for which we have found that are not relevant (De la Torre et al. 2021, Feng et al. 2022, Brishty et al.  2022).

Reviewer 2 Report

Krstulović et al. designed, synthesized 7-chloro-4-aminoquinoline-benzimidazole-based compounds and studied their antiproliferative properties using different cell lines. Additionally, the molecular docking approach was also applied to predict the binding mode of the selected Hit candidates against a particular drug target. Overall, this study is well-written and organized and can be considered for publication after a few modifications. My suggestions to authors are:

In section 2.2.1 what criteria was used for the selection of cell lines.

Table Indicates 5D and 5E displayed highest sensitivity among other candidates but on the other hand they were found active against the normal cell line right. Do authors try to optimize these compounds? Also, in the text, please explain more about why 12D was preferred over these compounds for easy understanding.

Section 2.3 What is MLOGP please explain this in the text as well. Why 5d was not selected as best candidate, from Table 2 I have not found a significant difference in pharmacokinetic properties.

SWISSADME provides drug-likeness properties only to predict ADMET properties tools like PKCSM or ADMET-SAR would be more useful.

Section 2.4. Please provide the rationale behind the selection of drug target. How SRC was selected. It would be great if reverse docking approach was considered to identify the best target for ligand.

In Table3 please remove pose number. And In Table 4 provide the H-bond distance as well

Figure 5 and 6 please uniform the ligand color.

Section 3.4 How the SRC PDB was selected: DFG IN/OUT explain. Which docking tool was used in calculation it not clear.

In conclusion line number 848 check the space

Author Response

Dear Reviewer 2,

We would like to thank you for your valuable comments and suggestions that improved our manuscript.

Our responses to your comments and questions:

  1. In section 2.2.1 what criteria was used for the selection of cell lines

Answer: The screening is performed with established tumour cell lines which were selected based on different origin and cellular sensitivity to chemotherapeutics.

Table Indicates 5D and 5E displayed highest sensitivity among other candidates but on the other hand they were found active against the normal cell line right. Do authors try to optimize these compounds?

Answer: Yes, we have synthesized similar compounds that will retain activity but will have a higher selectivity. These compounds will be presented in our future publications.

  1. Also, in the text, please explain more about why 12D was preferred over these compounds for easy understanding.

Answer: We have explained in subchapter 2.3. Absorption, distribution, metabolism, excretion (ADME), and toxicity properties (Page 12 ) and in the Chapter 4. Conclusions (Page 26).

  1. Section 2.3 What is MLOGP please explain this in the text as well.

Answer: MLOGP has been additionally explained in the text according to the reviewer's suggestion.

  1. Why 5d was not selected as best candidate, from Table 2 I have not found a significant difference in pharmacokinetic properties.

Answer: Although compound 5d exhibited strong inhibitory potential against the growth of all tested cancer cell line sit is not the best candidate since this compound proved to be toxic to the normal, MDCK1 cells (GI50 = 20.4 μM).

  1. SWISSADME provides drug-likeness properties only to predict ADMET properties tools like PKCSM or ADMET-SAR would be more useful.

Answer: Pharmacokinetic and toxicity properties have been additionally calculated and presented in the revised paper according to the reviewer's suggestion. We have calculated pharmacokinetic behaviour using the suggested program pkCSM, results are presented in Table 3 and discussed.

  1. Section 2.4. Please provide the rationale behind the selection of drug target. How SRC was selected. It would be great if reverse docking approach was considered to identify the best target for ligand.

Answer: Nonreceptor protein-tyrosine kinase (c-Src) is a protooncogene that plays key roles in cell morphology, motility, proliferation, and survival. Numerous studies have shown evidence of an association between c-Src kinases and leukemia. It was demonstrated that c-Src kinases play an important role in both assembly and disassembly of tight junctions in CaCo-2 and MDCK1 cell monolayers. These tight junctions mediate adhesion and communication between adjoining cells and provide a barrier within the membrane. The oxidative stress-induced disruption of tight junction is mediated by the activation of c-Src [Additionally explanation was introduced in discussion. I our previously paper we have used protein-tyrosine kinase as anticancer drug target as well:

  • Rastija, V.; Jukić, M.; Opačak-Bernardi, T.; Krstulović, L.; Stolić, I.; Glavaš-Obrovac, Lj.; Bajić, M. Investigation of the structural and physicochemical requirements of quinoline-arylamidine hybrids for the growth inhibition of K562 and Raji leukemia cells. Turk. J. Chem. 2019, 43, 251-265, doi:10.3906/kim-1807-61.
  • El-Sharkawy, E.R., Almalki, F., Ben Hadda T.; Rastija, V.; Lafridi, H.; Zgou, H. DFT calculations and POM analyses of cytotoxicity of some flavonoids from aerial parts of Cupressus sempervirens: Docking and Identification of Pharmacophore Sites. Bioorg. Chem. 2020, 100, 103850, doi:10.1016/j.bioorg.2020.103850.
  • Molnar, Maja; Lončarić, Melita; Opačak-Bernardi, Teuta; Glavaš-Obrovac, Ljubica; Rastija, Vesna, Rhodanine derivatives as anticancer agents - QSAR and molecular docking studies // Anti-Cancer Agents in Medicinal Chemistry (2022), (online) DOI: 10.2174/1871520623666221027094856

  1. In Table 3 please remove pose number. And In Table 4 provide the H-bond distance as well

Answer: The pose number was removed from the Table 4 (before Table 3). In the Table 4 was provided the H-bond distances).

  1. Figure 5 and 6 please uniform the ligand color.

Answer: We have changed Figure 5 informing the ligand colour and type of the receptor surface.

Section 3.4 How the SRC PDB was selected: DFG IN/OUT explain. Which docking tool was used in calculation it not clear.

Answer: As we mentioned in the manuscript, Crystal coordinates of the c-Src (PDB ID: 3G6H) in the complex with DSA compound were downloaded from the Protein Data Bank (PDB, https://www.rcsb.org/). We have also mentioned that we used the i GEMDOCK (BioXGEM, Taiwan) as a docking tool. DFG IN/OUT was additionally explained in section 2.4.

 In conclusion line number 848 check the space

Answer: We have checked the space.

Reviewer 3 Report

The article “Novel 7-Chloro-4-Aminoquinoline-Benzimidazole Hybrids as Inhibitors of Cancer Cells Growth. Synthesis, Antiproliferative Activity, In Silico ADME Predictions, and Docking” describes the antitumor activity of 14 new quinoline-benzimidazole hybrids towards different cancer cell lines, giving further insights about their ADME, docking and mechanism.

The manuscript is well written and organized, presenting interesting data and results for the field. Hence, it is suitable for the publication after the following minor revisions:

- A clear structure-activity relationship (SAR) is missing. Authors must add this important section.

- At the end of line 119, authors should try to explain why they did not yield the chlorine derivative

- In paragraph 2.2.1, specify the tumour subtypes to which each tested cell line belong

- For greater clarity, uniform the units of measurement for GI50 using µM

- References must be improved in the introduction. In particular, add examples of recent research works to prove the concrete efforts of medicinal chemists “to introduce new anti-cancer entities” (line 41-42): EJMC 243 (2022) 114744 - DOI: 10.1016/j.ejmech.2022.114744; Journal of Medicinal Chemistry 2022, 65, 21, 14891-14915 - DOI: 10.1021/acs.jmedchem.2c01431; Drug Dev Res. 2022;83:1331–1341 DOI: 10.1002/ddr.21962; Journal of Medicinal Chemistry  2022, 65, 22, 15165-15173 - DOI: 10.1021/acs.jmedchem.2c01010; EJMC 237:114399 DOI: 10.1016/j.ejmech.2022.114399

Author Response

Dear Reviewer 3,

We would like to thank you for your valuable comments and suggestions that improved our manuscript.

Our responses to your comments and questions:

  1. A clear structure-activity relationship (SAR) is missing. Authors must add this important section.

Answer: We have additionally explained the SAR in section 2.2.1. (Page 5, Lines 165 – 171) and added a new Figure (now Figure 2. Figure 2. Structure activity relationship for antiproliferative activity of the 7-chloro-4-aminoquinoline-benzimidazole compounds; Page 5)

  1. At the end of line 119, authors should try to explain why they did not yield the chlorine derivative

We added an explanation in the text (Page 4, Line 133)

  1. In paragraph 2.2.1, specify the tumour subtypes to which each tested cell line belong

Answer: The tumour subtypes are specified. (Page 5, Line 164-171): HeLa (human cervical adenocarcinoma), CaCo-2 (human colorectal adenocarcinoma), MCF-7 (human breast adenocarcinoma), CCRF-CEM (human acute lymphoblastic leukemia), Hut78 (T-cell lymphoma), THP-1 (acute monocytic leukemia), Burkitt lymphoma (Raji) and MDCK1 (Madine-Darby canine kidney)

  1. For greater clarity, uniform the units of measurement for GI50 using µM

Answer: Changed according suggestion

  1. References must be improved in the introduction. In particular, add examples of recent research works to prove the concrete efforts of medicinal chemists “to introduce new anti-cancer entities” (line 41-42): EJMC 243 (2022) 114744 - DOI: 10.1016/j.ejmech.2022.114744; Journal of Medicinal Chemistry 2022, 65, 21, 14891-14915 - DOI: 10.1021/acs.jmedchem.2c01431; Drug Dev Res. 2022;83:1331–1341 DOI: 10.1002/ddr.21962; Journal of Medicinal Chemistry  2022, 65, 22, 15165-15173 - DOI: 10.1021/acs.jmedchem.2c01010; EJMC 237:114399 DOI: 10.1016/j.ejmech.2022.114399

Answer: we have added examples of recent research on the efforts “to introduce new anti-cancer entities”(Mao et al. 2022, Labozzeta et al. 2022, Moreno et al. 2022), we have also added one reference for the part that is talking about hybrid drugs (Szumilak et al. 2021). We have removed three references from the introduction part, for which we have found that are not relevant (De la Torre et al. 2021, Feng et al. 2022, Brishty et al.  2022).

Round 2

Reviewer 1 Report

The authors have addressed all of my points and the modified version reflects all required changes.